# Microhomology-mediated circular DNA formation from oligonucleosomal fragments during spermatogenesis

Jun Hu[1†], Zhe Zhang[2†], Sai Xiao[3], Yalei Cao[2], Yinghong Chen[3], Jiaming Weng[2], Hui Jiang[2,4]*, Wei Li[3,5]*, Jia-Yu Chen[1,6]*, Chao Liu[3,5]*

[1]State Key Laboratory of Pharmaceutical Biotechnology, School of Life Sciences, Chemistry and Biomedicine Innovation Center (ChemBIC), Department of Andrology, Nanjing Drum Tower Hospital, Nanjing University, Nanjing, China; [2]Department of Urology, Department of Reproductive Medicine Center, Peking University Third Hospital, Beijing, China; [3]Guangzhou Women and Children's Medical Center, Guangzhou Medical University, Guangzhou, China; [4]Department of Urology, Peking University First Hospital Institute of Urology, Beijing, China; [5]State Key Laboratory of Stem Cell and Reproductive Biology, Institute of Zoology, Stem Cell and Regenerative Medicine Innovation Institute, Chinese Academy of Science, Beijing, China; [6]Nanchuang (Jiangsu) Institute of Chemistry and Health, Nanjing, China

*For correspondence:
jianghui55@163.com (HJ);
leways@gwcmc.org (WL);
jiayuchen@nju.edu.cn (J-YC);
liuchao@gwcmc.org (CL)

[†]These authors contributed equally to this work

Competing interest: The authors declare that no competing interests exist.

**Abstract** The landscape of extrachromosomal circular DNA (eccDNA) during mammalian spermatogenesis, as well as the biogenesis mechanism, remains to be explored. Here, we revealed widespread eccDNA formation in human sperms and mouse spermatogenesis. We noted that germline eccDNAs are derived from oligonucleosomal DNA fragmentation in cells likely undergoing cell death, providing a potential new way for quality assessment of human sperms. Interestingly, small-sized eccDNAs are associated with euchromatin, while large-sized ones are preferentially generated from heterochromatin. By comparing sperm eccDNAs with meiotic recombination hotspots and structural variations, we found that they are barely associated with de novo germline deletions. We further developed a bioinformatics pipeline to achieve nucleotide-resolution eccDNA detection even with the presence of microhomologous sequences that interfere with precise breakpoint identification. Empowered by our method, we provided strong evidence to show that microhomology-mediated end joining is the major eccDNA biogenesis mechanism. Together, our results shed light on eccDNA biogenesis mechanism in mammalian germline cells.

## eLife assessment

This study provides **important** information on the biogenesis of eccDNAs during spermatogenesis. The data presented are **solid** and supportive of the concussion that eccDNAs in spermatogenic cells are not derived from miotic recombination hotspots but rather represent oligonucleosomal DNA fragments from apoptotic male germ cells, whose ends are ligated through microhomology-mediated end-joining. This work is of interest to researchers working on germ cell biology and cancer biology.

## Introduction

Apart from linear chromosome, DNA in circular form also exists in the nuclei of eukaryotes (***Noer et al., 2022***). Circular DNAs could be roughly classified into two groups based on their cell origins

(*Chiu et al., 2020*). The one exclusively present in cancerous cells is usually referred to as extrachromosomal DNA (eccDNA), which is of megabase long in average and plays key roles in tumorigenesis (*Wang et al., 2021b*). The ecDNA biogenesis is linked to 'episome model' (*Carroll et al., 1988*), chromothripsis (*Shoshani et al., 2021*), breakage-fusion-bridge (*Coquelle et al., 2002*), or translocation-deletion-amplification model (*Van Roy et al., 2006*). The other class found in somatic and germ cells is usually called extrachromosomal circular DNA (eccDNA). The size of eccDNA ranges from dozens of bases to hundreds of kilobases (*Paulsen et al., 2018*). Contrary to ecDNAs, the biogenesis mechanisms and biological functions of eccDNAs are relatively less experimentally characterized, and current studies show inconclusive or even contradictory results.

The genomic origins of eccDNAs have been extensively investigated in different cells and conditions with the application of Circle-seq and its refined derivatives (*Mehta et al., 2020*; *Møller et al., 2015*; *Wang et al., 2021a*), where eccDNAs are detected via rolling circle amplification and deep sequencing. While CpG islands (*Shibata et al., 2012*), gene-rich regions (*Møller et al., 2018*), and repeat elements, for example, LTR (long terminal repeat) (*Møller et al., 2015*), LINE-1 (*Dillon et al., 2015*), segmental duplication (*Mouakkad-Montoya et al., 2021*), or satellite DNA (*Mouakkad-Montoya et al., 2021*), are hotspots for eccDNA formation, others found that eccDNAs are nearly random with regard to genomic distribution (*Møller et al., 2020*), or even made opposite observations (*Henriksen et al., 2022*). Epigenomically, the overall higher GC content, the periodicities of dinucleotide, and eccDNA size convergently point to that nucleosome wrapping of DNA might contribute to the formation of small-sized eccDNAs (*Shibata et al., 2012*; *Wang et al., 2021a*), an intriguing starting point for mechanistic understanding of eccDNA origination. However, direct evidence for coincident positioning of eccDNAs and nucleosomes is still lacking, not to mention specific epigenetic marks on nucleosomes that are tightly associated with eccDNA formation.

EccDNAs are increased upon DNA damages (*Møller et al., 2015*; *Paulsen et al., 2021*), suggesting them as by-products of successive DNA repairs. Among diverse repair pathways, it was reported that eccDNA levels particularly depend on resection after double-strand DNA break (DSB) and repair by microhomology-mediated end joining (MMEJ) (*Paulsen et al., 2021*; *Wang et al., 2021a*). In further support of the involvement of MMEJ, microhomology is found around eccDNA breakpoints (*Lukaszewicz et al., 2021*; *Møller et al., 2015*). However, in all studies using short-read sequencing technologies, eccDNA breakpoints are mis-annotated if microhomologous sequences are present around due to their interference to precise breakpoint detection (see our main text; *Dillon et al., 2015*; *Henriksen et al., 2022*; *Kumar et al., 2017*; *Lv et al., 2022*; *Mann et al., 2022*; *Møller et al., 2018*; *Møller et al., 2020*; *Paulsen et al., 2019*; *Prada-Luengo et al., 2019*; *Shibata et al., 2012*; *Sin et al., 2020*; *Wang et al., 2021a*; *Zhang et al., 2021*), or the eccDNA identification does not depend on precise breakpoint detections at all (*Møller et al., 2015*; *Mouakkad-Montoya et al., 2021*). The contribution of microhomology to eccDNA generations thus needs to be revisited with precise mapping of breakpoints.

Alternative or additional mechanisms might be involved in germline eccDNA formation. During meiosis, two spatially closed break sites catalyzed by SPO11 at recombination hotspots may release eccDNAs accompanied by de novo deletions on linear chromosomes (*Lukaszewicz et al., 2021*). Consistently, germline microdeletions display similar sequence features with eccDNAs (*Shibata et al., 2012*). However, a recent study reported that the creation of germline eccDNAs negatively correlate with meiotic recombination rates (*Henriksen et al., 2022*). Therefore, it remains to be determined whether meiosis might significantly contribute to eccDNA biogenesis.

We envision that eccDNA landscape during spermatogenesis is ideal for clarifying the abovementioned issues and so better understand the biogenesis mechanisms and biological implications of eccDNAs. Only a small fraction of histones will survive from the histone-to-protamine transition in mature sperms, allowing us to more specifically correlate eccDNA origination with histones. Studying eccDNAs in germline cells rather than somatic cells could help reveal to what extent meiosis might contribute to eccDNA generation and de novo structural variations that can be passed to offspring. Therefore, in this study, we profiled eccDNAs via Circle-seq in human sperms and different developmental stages of mouse germ cells with an improved analysis pipeline to identify eccDNAs at nucleotide resolution. We conclude that germline eccDNAs are likely formed by microhomology-mediated ligation of nucleosome-protected fragments and barely contribute to de novo genomic deletions at meiotic recombination hotspots.

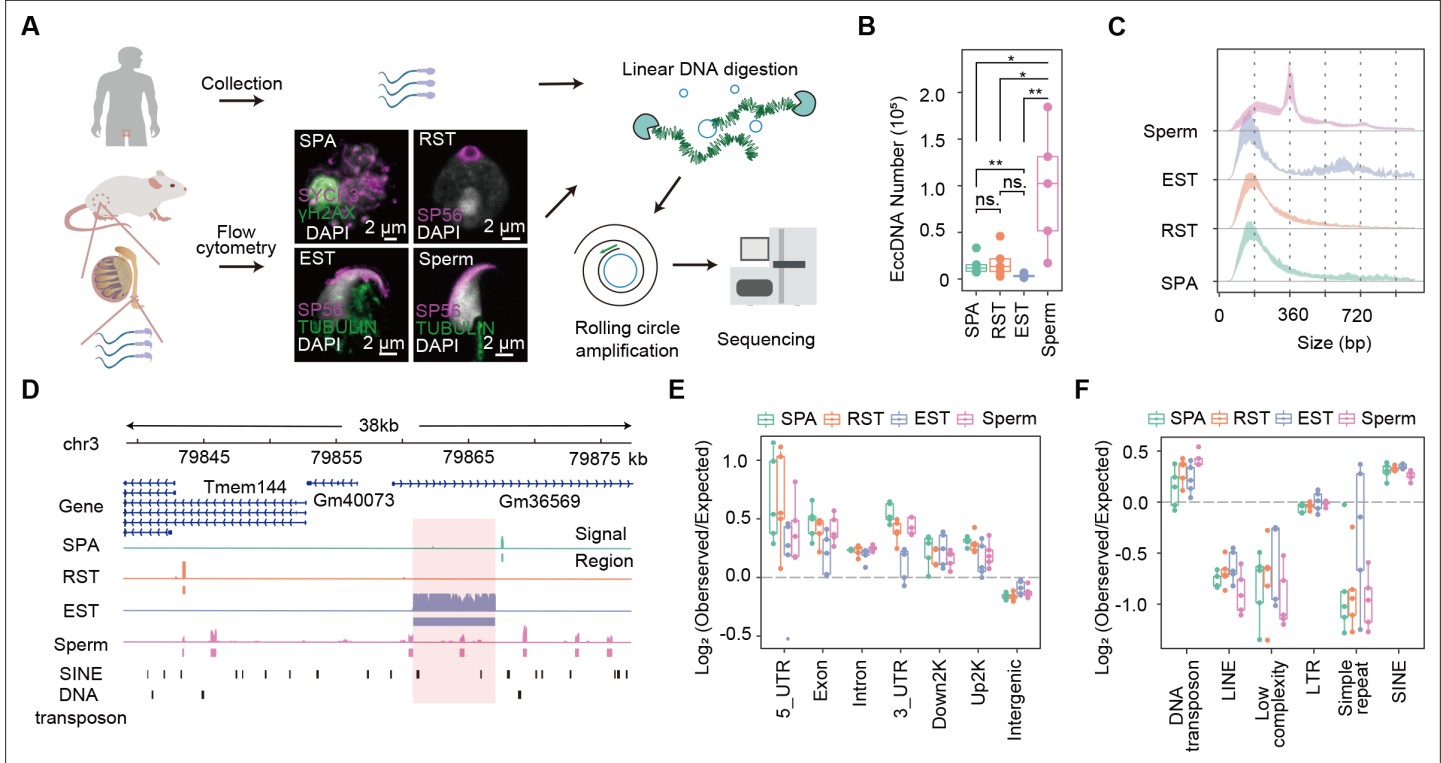

**Figure 1.** Overview of extrachromosomal circular DNA (eccDNA) formation during mouse spermatogenesis. (**A**) Schematic representation of Circle-seq in human sperm cells and mouse spermatocytes (SPA), round spermatids (RST), elongated spermatids (EST), and sperm cells validated with immunochemistry. SYCP3: a component of the synaptonemal complex; γH2AX: a marker for double-strand breaks; SP56: a marker for acrosome organelle; TUBULIN: structural component of manchette in EST and flagellum axoneme in sperm cells. (**B**) Number of eccDNAs detected in different cell types. *Two-sided *t*-test p-value<0.05; **two-sided *t*-test p-value<0.01. (**C**) Size distribution of eccDNAs during mouse spermatogenesis. Dotted lines indicate multiplies of 180 bp. (**D**) A representative genomic locus showing the gene annotation, Circle-seq signals, detected eccDNAs, and SINE and DNA repeat elements. Highlighted in red rectangle is a large-sized eccDNA. (**E**) Enrichment of eccDNAs at given genomic regions relative to randomly-selected control regions. (**F**) Enrichment of eccDNAs at given repeat elements relative to randomly selected control regions.

The online version of this article includes the following source data and figure supplement(s) for figure 1:

**Figure supplement 1.** Detection and characterization of extrachromosomal circular DNAs (eccDNAs).

**Figure supplement 2.** Quality control of the extrachromosomal circular DNA (eccDNA) isolation procedure.

**Figure supplement 2—source data 1.** The original gel images for *Figure 1—figure supplement 2C* showing PCR validation of three extrachromosomal circular DNAs (eccDNAs) using inward and outward PCR primers.

**Figure supplement 3.** Length distribution of extrachromosomal circular DNAs (eccDNAs) in different cells.

# Results

## Widespread eccDNA formation in human and mouse germline cells

Because it was reported that high content of eccDNAs existed in sperms (*Henriksen et al., 2022*), we examined the genome-wide eccDNA landscape in two human sperm samples by Circle-seq (*Møller et al., 2018*; *Møller et al., 2015*; *Møller et al., 2020*; see 'Materials and methods') and indeed found that there were widespread eccDNAs across the human genome (*Figure 1A*, *Figure 1—figure supplement 1A*). This motivated us to further investigate the biogenesis mechanism, particularly whether it might be linked to specific spermatogenesis processes. Given that it is ethically prohibited and technically challenging to collect pure spermatogenic cell types from human individuals, we therefore turned to use mouse model to study the eccDNA formation during spermatogenesis.

A series of cell divisions and morphological changes are involved in spermatogenesis, where spermatogonial stem cells develop into spermatocytes (SPA) via mitosis, and SPA then undergo meiosis to produce haploid round spermatids (RST), which will take a dramatic morphological change and chromatin compaction to produce elongated spermatids (EST) and finally matured sperms (*Hess*

*and Renato de Franca, 2008*; *ROOSEN-RUNGE, 1962*). We isolated SPA, RST, and EST using flow cytometry and collected sperms from mouse cauda epididymis (*Hayama et al., 2016*) for subsequent Circle-seq (*Møller et al., 2018*; *Møller et al., 2015*; *Møller et al., 2020*; see 'Materials and methods'). All four cell types were validated with known markers and cell morphology (*Figure 1A*). EccDNA isolation procedures were validated by a high ratio of an exogenous circular DNA (pUC19) to a linear DNA locus (H19 gene) (*Figure 1—figure supplement 2A*), and the low abundance of mitochondria DNA that was supposed to be cleaved by PacI and degraded by exonuclease (*Figure 1—figure supplement 2B*). To account for sample variations, up to five biological replicates and ~150 million reads for each cell type were sequenced for eccDNA detection. From ~1500 to ~180,000 high-confidence eccDNAs were identified, suggesting widespread circular DNA formation during mouse spermatogenesis (*Figure 1B*; see 'Materials and methods'). Some randomly selected eccDNAs were validated with PCR using outward primers (*Figure 1—figure supplement 2C*). The reproducible rate of eccDNAs with 50% reciprocal overlap between biological replicates was only ~2.4% in average, a level comparable to previous studies (*Henriksen et al., 2022*; *Møller et al., 2018*; *Figure 1—figure supplement 1B*). As noted earlier (*Møller et al., 2018*), the detected eccDNAs seemed not saturated (*Figure 1—figure supplement 1C*; see 'Discussion' and 'Materials and methods'), which might underlie the observed low reproducibility. Nevertheless, principal component analysis suggested that the within-group similarity was marginally higher than the between-group similarity (*Figure 1—figure supplement 1D*), allowing investigation of stage-specific eccDNA features during mouse spermatogenesis.

The detected germline eccDNAs verified known genomic features of eccDNAs. First, the natural size distribution of eccDNA is usually distorted in Circle-seq as smaller eccDNAs tend to be overrepresented in rolling circle amplification (*Mohsen and Kool, 2016*; *Møller et al., 2018*; *Møller et al., 2015*). As expected, the detected eccDNA population was dominated by small-sized eccDNAs, most of which were ~180 bp or ~360 bp long (*Figure 1C*). However, eccDNA size could occasionally reach to several kilobases and even tens of kilobases (*Figure 1D*). Second, eccDNAs from different cell types were all enriched at gene-rich regions, especially 5'UTR (*Figure 1E*, *Figure 1—figure supplement 1E*), corroborating the reported association between eccDNA frequency and gene density in somatic cells (*Dillon et al., 2015*; *Shibata et al., 2012*). ccDNAs were also highly associated with SINE but not LINE elements (*Figure 1F*), and quantitative analysis revealed that eccDNA biogenesis was positively correlated with SINE density (*Figure 1—figure supplement 1F*), but negatively correlated with LINE density (*Figure 1—figure supplement 1G*). Given that SINE and LINE elements function to orchestrate chromosomes into gene-rich A compartment and gene-poor B compartment, respectively (*Lu et al., 2021*), the positive correlation between eccDNAs and SINE elements might further support that eccDNAs are overall highly associated with the gene-rich regions. Interestingly, we also noticed a strong association between eccDNAs and DNA transposons (*Figure 1F*), suggesting that DNA transposons might get circularized rather than or in addition to reintegrated into the genome, an interesting possibility awaiting further investigations. Altogether, the genome-wide eccDNA landscape during mouse spermatogenesis allows us to further study the biogenesis mechanism and function of eccDNAs.

## High eccDNA load and periodic eccDNA size distribution in mouse sperm cells

Notably, sperm cells had 97,372 eccDNAs detected in average, a number significantly higher than those in SPA (15,246), RST (18,426), and EST (3591) cells (*Figure 1B*). SPA cells did not show higher eccDNA numbers (*Figure 1B*), suggesting that meiosis does not seem to contribute significantly to eccDNA biogenesis. Since the same amount of eccDNAs (10 ng) was used for library construction and all samples were sequenced in comparable and sufficiently-deep depth, it suggests that eccDNA species in sperm cells has higher complexity. However, the higher starting cell number for sperm cells might account for the larger diversity of sperm eccDNA species (see 'Discussion' and 'Materials and methods'); otherwise, it would be interesting to explore any specific features of sperm cells underlying the higher load of eccDNAs.

In contrast to SPA, EST, and RST eccDNAs showing the unimodal distribution that was centered at ~180 bp, sperm-derived eccDNAs showed a multimodal distribution with a pronounced periodicity of ~180 bp (*Figure 1C*), which was readily seen in individual samples (*Figure 1—figure supplement 3*). Given that each nucleosome consists of 147 bp DNA wrapping itself around a histone core,

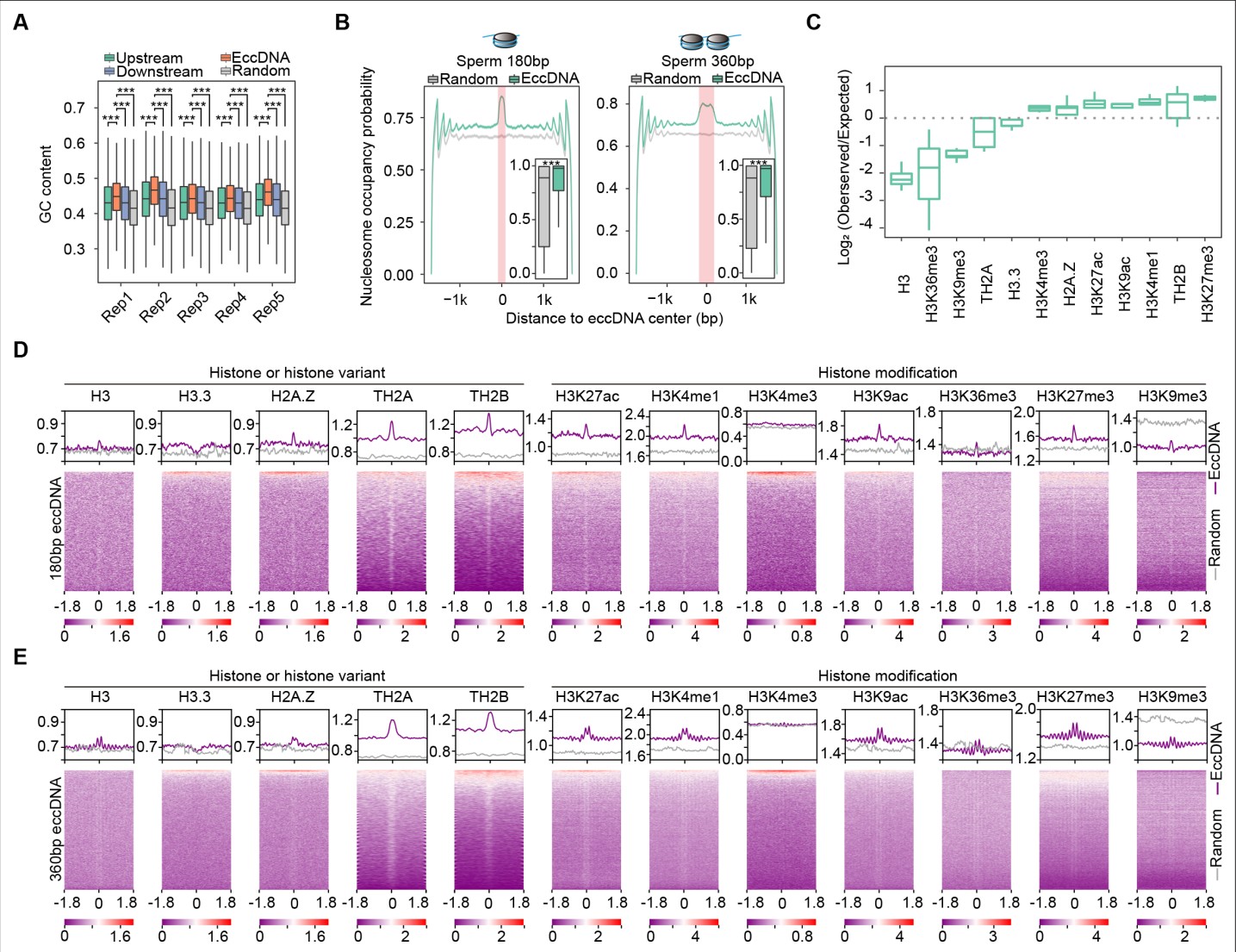

**Figure 2.** Association between sperm extrachromosomal circular DNAs (eccDNAs) and nucleosome positioning. (**A**) GC contents of sperm eccDNAs, regions upstream and downstream of eccDNAs, and randomly selected length-matched control regions. ***Two-sided *Wilcoxon* test p-value<0.001. (**B**) Predicted probability of nucleosome occupancy for eccDNA and randomly selected length-matched control regions (highlighted by red-shaded area), and surrounding regions. Boxplots showing the probability distribution of individual eccDNAs and control regions. ***Two-sided *Wilcoxon* test p-value<0.001. (**C**) Enrichment of eccDNAs at different histones and histone modifications. (**D, E**) ChIP-seq signal distribution at [–1.8 kb, +1.8 kb] of the centers of ~180 bp (**D**) and ~360 bp (**E**) eccDNAs. ChIP-seq signals quantified as reads density are color-coded below heatmaps.

the ~180-bp-long fragments likely corresponded to histone core region plus ~20–30 bp linker regions, as observed in apoptotic cells (***Matassov et al., 2004***). Although the identified eccDNAs in all spermatogenesis stages were likely related to nucleosomes, the different modes of size distribution might be due to distinct nucleosome compositions and structures between sperm and other spermatogenic cells.

## Mouse sperm eccDNAs come from DNA fragments protected by histones

Only a small fraction of histones will be retained in mouse sperm cells after histone-to-protamine transition (***Torres-Flores and Hernández-Hernández, 2020***), permitting us to more specifically correlate eccDNAs with histones. We were therefore motivated to see whether the detected eccDNAs were derived from the retained histones in mature sperm cells. We noted that sperm eccDNAs had higher GC content than surrounding regions as well as control regions randomly selected across the genome

(*Figure 2A*), resembling the sequence feature of nucleosome-protected DNA fragments. Consistently, sequence-based prediction revealed significantly higher nucleosome occupancy probability for ~180 bp (from 175 bp to 185 bp) and ~360 bp (from 355 bp to 365 bp) sperm eccDNA regions (*Figure 2B*; see 'Materials and methods'). A small dip was observed at the center of ~360 bp eccDNA regions, which likely corresponded to the linker region between two nucleosomes (*Figure 2B*, right).

It is a common practice to reuse publicly available genomics data generated in the same cell types for integrative analysis. Taking advantage of public ChIP-seq data for histones and their modifications in mouse sperm cells (*Jung et al., 2019*; *Jung et al., 2017*; *Singh and Parte, 2021*), we found that eccDNAs were significantly enriched with certain histone variants and modifications (*Figure 2C*), and 7.46% of sperm eccDNAs in total were intersected with at least one ChIP-seq peaks. Considering that histones occasionally retained in sperms might not generate strong ChIP-seq signals exceeding the peak calling cutoff, a meta-gene analysis of ChIP-seq signals at and around sperm eccDNA regions will likely provide more insights. Interestingly, enrichment of H3 histone and H2A.Z, TH2A, and TH2B histone variants but depletion of H3.3 variant was observed at ~180 bp sperm eccDNA regions (*Figure 2D*). These eccDNAs also showed strong associations with H3K27ac, H3K4me1, H3K9ac, and H3K27me3 modifications; however, no enrichment was seen for H3K4me3, and H3K36me3 and H3K9me3 signals were comparable with or even lower than randomly selected regions as control (*Figure 2D*).

We next examined ~360 bp sperm eccDNAs, which supposedly correspond to two nucleosomes and made similar observations. Centers of ~360 bp eccDNAs were well positioned between two adjacent nucleosomes consisting of H3 histone and H2A.Z histone variants, and H3K27ac, H3K4me1, H3K9ac, and H3K27me3 histone modifications (*Figure 2E*). Similar to ~180 bp eccDNAs, ~360 bp eccDNAs did not show association with H3.3 or H3K4me3, or stronger association than randomly selected regions with H3K36me3 and H3K9me3 either (*Figure 2E*). Although H3.3 variant coincides with active transcription, it is also well known for its localization at heterochromatin region and its roles in promoting heterochromatin formation by inhibiting H3K9/K36 histone demethylase (*Udugama et al., 2022*). Together, euchromatin is generally more preferred than heterochromatin for eccDNA biogenesis, which is consistent with the enrichment of sperm eccDNAs at gene-rich regions (*Figure 1D*).

## Large-sized eccDNAs are preferentially generated from heterochromatin regions

Intriguingly, periodic distribution of nucleosomes, for example, those marked with H3K27me3, was observed for ~360 bp but not for ~180 bp eccDNAs, indicating that eccDNAs from di-nucleosomes but not mono-nucleosomes preferentially originate from well-positioned nucleosome arrays

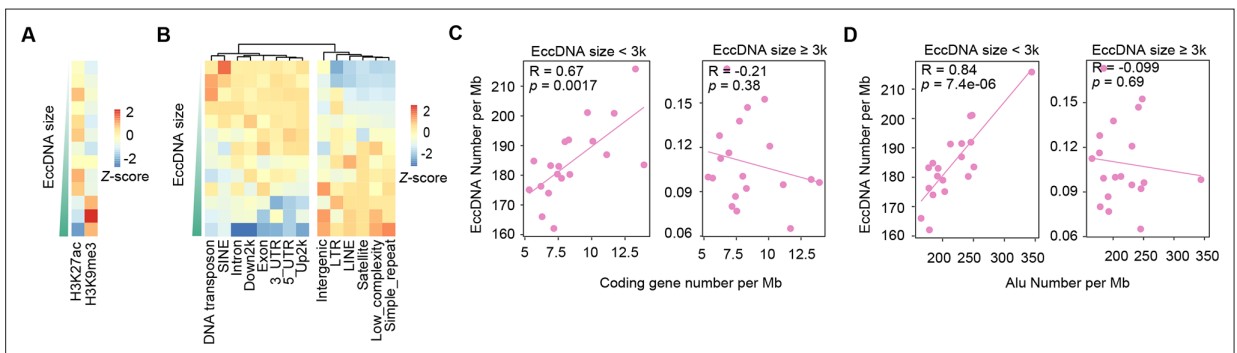

**Figure 3.** Large-sized extrachromosomal circular DNAs (eccDNAs) are preferentially generated from heterochromatin regions. (**A**) Distribution at H3K27ac- and H3K9me3-marked regions for eccDNAs of different sizes. (**B**) Distribution at different genomic regions for eccDNAs of different sizes. (**C**) Number of small (<3 kb) vs. large (≥3 kb) eccDNAs per Mb as a function of gene number per Mb. *Pearson* correlation coefficients and two-sided *t*-test p-values are indicated. (**D**) Number of small (<3 kb) vs. large (≥3 kb) eccDNAs per Mb as a function of Alu number per Mb. *Pearson* correlation coefficients and two-sided *t*-test p-values are indicated.

The online version of this article includes the following figure supplement(s) for figure 3:

**Figure supplement 1.** Correlation between the density of small-sized (**A**) vs. large-sized (**B**) extrachromosomal circular DNAs (eccDNAs) and the meiotic recombination rate (*Jensen-Seaman et al., 2004*).

(*Figure 2E*). We were further prompted to ask whether eccDNAs of different sizes are originated from different genomic regions. Indeed, small-sized eccDNAs (<3 kb) were more enriched at H3K27ac-marked euchromatin regions, while large-sized ones (≥3 kb) at H3K9me3-marked heterochromatin regions (*Figure 3A*). Accordingly, small-sized eccDNAs were generally more associated with genic regions, while large-sized ones with non-genic regions (*Figure 3B*). Since sperm eccDNAs in this study were dominantly small-sized ones (*Figure 1C*), strong enrichment of eccDNAs at genic regions was observed (*Figure 1E*). However, strong depletion at genic regions was reported for human sperm eccDNAs in a recent study (*Henriksen et al., 2022*). Close inspection suggests that the discrepancy is partially reconciled in the light of two eccDNA groups of different sizes. Henriksen et al. studied eccDNAs with the size largely ranging from ~3 kb to 50 kb (*Henriksen et al., 2022*), rather than small-sized ones reported by us and many others (*Dillon et al., 2015*; *Møller et al., 2018*; *Møller et al., 2020*; *Paulsen et al., 2019*; *Shibata et al., 2012*; *Wang et al., 2021a*). This was why we chose 3 kb as the cutoff to separate eccDNAs into small- and large-sized categories. In support of this notion, the large-sized sperm eccDNAs detected in this study displayed a weak negative correlation with gene density or Alu elements (*Figure 3C and D*). Altogether, compared to euchromatin regions, heterochromatin regions are probably too condensed to be fragmented into smaller pieces for small-sized eccDNA formation.

## Germline eccDNAs as apoptotic products are not associated with meiotic recombination hotspots

The observed association between eccDNA and oligonucleosomal DNA fragmentation (*Figure 2*) is a typical feature of cell death. The spontaneous death of germ cells has been observed during the normal spermatogenesis (*Liu et al., 2017*; *Shaha et al., 2010*; *Weinbauer et al., 2001*; *Young et al., 2001*); however, it is still debatable whether spermatids and sperm can undergo apoptosis (*Lachaud et al., 2004*). Thus, sperm-derived eccDNAs might be associated with apoptosis (if exists) or unprogrammed cell death of germ cells during the spermatogenesis (see also 'Discussion'). In support of this hypothesis, all features associated with mouse germline eccDNAs identified in this study (*Figure 1C, E, and F*) closely matched with those of eccDNAs whose generation is dependent on apoptotic DNA fragmentation (*Figure 4—figure supplement 1*; *Wang et al., 2021a*).

During meiosis, two spatially closed cleavage sites catalyzed by SPO11 at recombination hotspots could release eccDNAs and generate de novo genomic deletions (*Lukaszewicz et al., 2021*), which, if transmitted to offspring, might contribute to structural variations within population. Since most sperm eccDNAs likely result from oligonucleosomal DNA fragments in sperm cells undergoing cell death (*Figures 2 and 3*) and SPA cells undergoing meiosis does give rise to more eccDNAs than other cells (*Figure 1B*), meiotic recombination is unlikely the major mechanism for germline eccDNA generation. To test this hypothesis, we first investigated to what extent eccDNA breakpoints well correspond to recombination hotspots defined as SPO11 or PRDM9 binding sites (*Alleva et al., 2021*; *Lange et al., 2016*). We noted that there was only a small number of eccDNAs with both breakpoints located in one recombination hotspot or two different hotspots (*Figure 4A*). These eccDNAs only constituted <0.15% (or <350) of mouse germline eccDNAs, suggesting a very low level of coincidence between eccDNA generation and meiotic recombination (*Figure 4A*). Consistently, only dozens of, or a few hundred eccDNAs in mouse germline cells coincided with known genomic deletions within mouse population (*Figure 4B*). Altogether, germline eccDNAs are likely apoptotic products that are not associated with meiotic recombination hotspots and heritable genomic deletions.

## Microhomology-directed ligation is the major biogenesis mechanism of germline eccDNAs

We therefore further explored how nucleosome-protected DNA fragments get circularized into eccDNAs. As suggested by previous studies, MMEJ is implicated in eccDNA biogenesis (*Lukaszewicz et al., 2021*; *Møller et al., 2015*; *Paulsen et al., 2021*; *Wang et al., 2021a*). The precise distribution of microhomologous sequences relative to eccDNA breakpoints will help better understand how and to what extent MMEJ might contribute to eccDNA biogenesis. However, we noted that the presence of microhomologous sequences will hinder precision eccDNA breakpoint identification (*Figure 4C*), which is not well dealt with by existing methods for eccDNA detection, including ECCsplorer (*Mann et al., 2022*), Circle_finder (*Kumar et al., 2017*), Circle_Map (*Prada-Luengo et al., 2019*), and

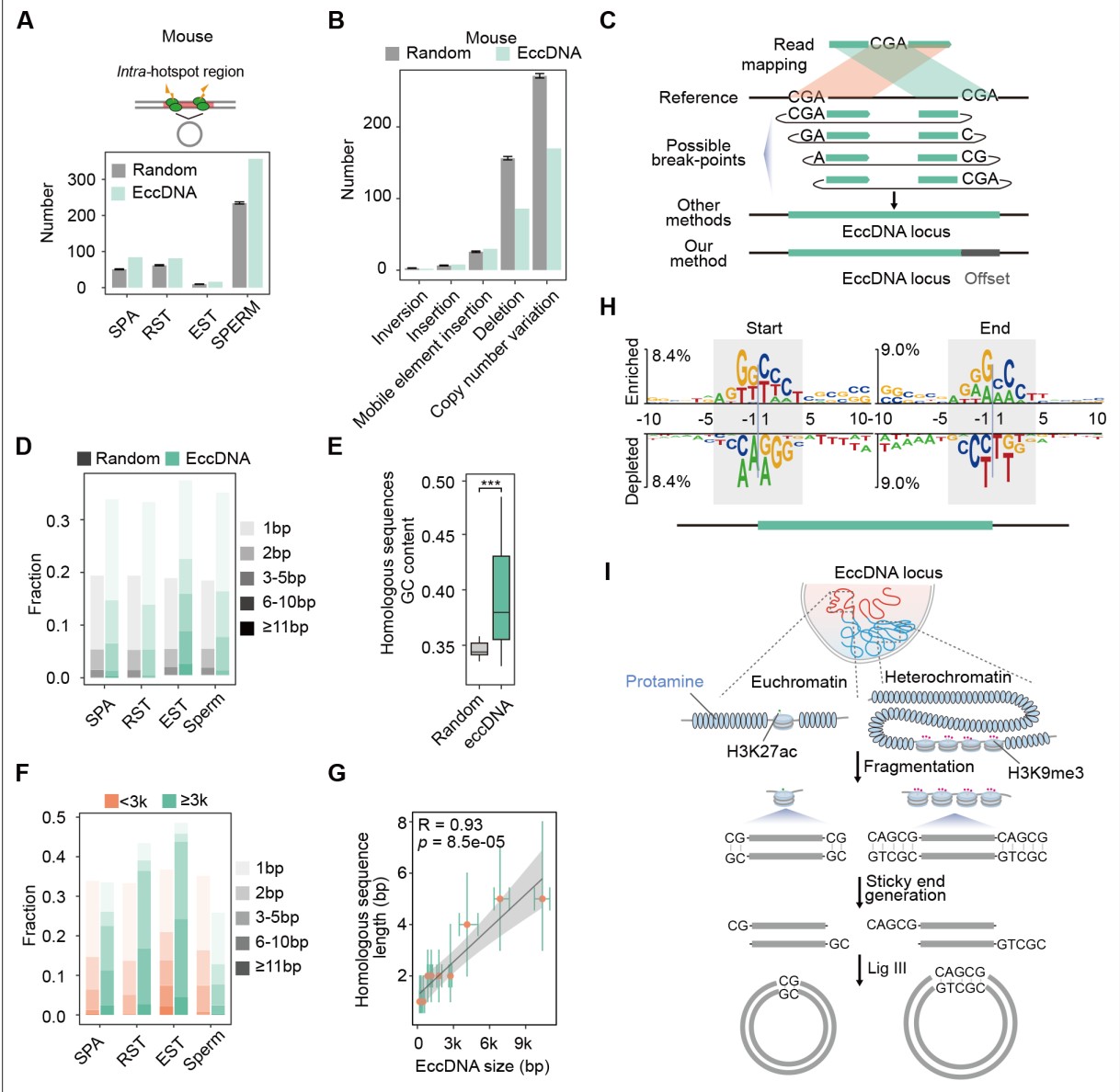

**Figure 4.** Microhomology-directed ligation accounts for emergence of most extrachromosomal circular DNAs (eccDNAs). (**A**) Numbers of eccDNAs or randomly selected control regions overlapped with recombination hotspots in mouse. (**B**) Shown are numbers of mouse sperm eccDNAs or randomly selected control regions having 95% reciprocal overlap with different types of structural variations. (**C**) Illustrated are how an eccDNA with homologous sequences (CGA) at two ends is identified from short-read sequencing data by our methods vs. other methods. (**D**) Percentages of homologous sequences of different lengths (coded by different color saturation levels) are shown for eccDNAs and randomly selected control regions. (**E**) GC content of homologous sequences and randomly selected control regions. (**F**) Percentages of homologous sequences of different lengths (coded by different color saturation levels) are shown for small-sized (<3 kb) and large-sized (≥3 kb) eccDNAs. (**G**) Length of microhomologous sequences as a function of the eccDNA size. Data points are shown as median plus lower (25%) and upper (75%) quartiles. The shaded area is 95% confidence interval of linear regression line. *Pearson* correlation coefficient and two-sided *t*-test p-value are indicated. (**H**) Sequencing motif analysis for ±10 bp leftmost left ends and ±10 bp leftmost right ends of eccDNAs with no perfectly matched homologous sequences observed. (**I**) Model for microhomology-mediated end joining (MMEJ)-directed eccDNA biogenesis.

The online version of this article includes the following figure supplement(s) for figure 4:

**Figure supplement 1.** High similarity between sperm extrachromosomal circular DNAs (eccDNAs) detected in this study and those from apoptotic DNA fragmentation reported previously.

**Figure supplement 2.** Evaluation of our nucleotide-resolution extrachromosomal circular DNA (eccDNA) detection method.

ecc_finder (*Zhang et al., 2021*; *Figure 4—figure supplement 2A*). Short sequencing reads spanning eccDNA breakpoints will be mapped to the genome as split reads, with its first part mapped to the right end of eccDNA, and the second part to the left end. If the sequence in front of the left eccDNA end is homologous to the right eccDNA end, or if the sequence following the right eccDNA end is homologous to the left eccDNA end, the homologous regions will be included in both parts of split reads to reach to a maximal length of matches, and many existing methods will mistake the eccDNA plus two homologous regions as the whole eccDNA region (*Figure 4C*). Being aware of it, we developed a base-resolution method for eccDNA identification on the basis of previous efforts (*Figure 4—figure supplement 2B*; *Kumar et al., 2017*; *Møller et al., 2018*). When homologous sequences are present, we record the coordinates of the leftmost form of eccDNA and an offset corresponding to the length of homologous sequences to represent all possible eccDNA variants (*Figure 4C*). Similar to ECCsplorer (*Mann et al., 2022*), Circle_finder (*Kumar et al., 2017*), Circle_Map (*Prada-Luengo et al., 2019*), and ecc_finder (*Zhang et al., 2021*), our method was not designed to identity eccDNAs that encompass multiple gene loci.

We evaluated the performance of our method in comparison with existing methods. Firstly, we simulated paired-end reads derived from a set of eccDNAs with homologous sequences around breakpoints and employed all methods for eccDNA identification (see 'Materials and methods'). In total, 97.9, 97.9, 97.4, 95.3, and 91.1% eccDNA regions could be detected by our method, Circle_Map, Circle_finder, ecc_finder, and ECCsplorer, respectively (*Figure 4—figure supplement 2C*). This result suggests that our method has comparable performance with existing methods in detecting eccDNA regions. Moreover, our method could faithfully assign breakpoints with 97.4% accuracy, in contrast to no more than 15% by other methods (*Figure 4—figure supplement 2D*). Secondly, we applied all methods on one dataset generated in this study. Again, our method had comparable sensitivity and specificity with existing methods (especially Circle_finder and Circle_Map) in detecting eccDNA regions (*Figure 4—figure supplement 2E*). At least 60% of eccDNAs with homologous sequences were misannotated by ECCsplorer, ecc_finder, Circle_finder, and Circle_Map, respectively (*Figure 4—figure supplement 2A and F*). Overall, our method shows a high efficiency and accuracy in precise eccDNA detection.

In contrast to simulated controls (15%), more than one-third of eccDNAs had ≥1 bp homologous sequences, most of which were shorter than 5 bp (*Figure 4D*), suggesting the involvement of MMEJ in eccDNA biogenesis. The GC content of homologous sequences was higher than that of simulated control regions, permitting stronger base-pairing for efficient MMEJ (*Figure 4E*). Considering that two free-ends of long DNA fragments might be not as spatially close as those of short DNA fragments, formation of longer eccDNA should more rely on longer homologous sequences for stable base-pairing. Indeed, large-sized eccDNAs in SPA, RST, and EST cells did show higher percentage of ≥2 bp homology than small-sized eccDNAs, and large-sized eccDNAs in sperm cells showed higher percentage of >5 bp homology (*Figure 4F*). A significant positive correlation between lengths of homologous sequences and eccDNA sizes was observed (*Figure 4G*). We further reasoned that for the remaining two-thirds of eccDNAs that were lack of perfectly matched homologous sequences, imperfect homologous sequences might be present. Accordingly, we noted the same sequence motifs between eccDNA starts and sequences following eccDNA ends, and between eccDNA ends and sequences in front of eccDNA starts (*Figure 4H*). Similar observations have been made also by others before (*Sin et al., 2020*); however, they failed to precisely locate the homologous sequences relative to eccDNA breakpoints. We propose that sticky ends of DNA fragments with homologous sequences might base pair with each other, and then be ligated by DNA ligase, for example, DNA ligase III (*Wang et al., 2021a*), to form eccDNAs (*Figure 4I*). In sum, among all proposed mechanisms (*Chiu et al., 2020*; *Dillon et al., 2015*; *Møller et al., 2015*; *Paulsen et al., 2021*; *Sin et al., 2020*), MMEJ-mediated ligation accounts for emergence of most eccDNAs at least in germline cells.

## The eccDNA biogenesis mechanism is conserved in somatic tissues and in human

Upon revealing the major biogenesis mechanism of mouse germline eccDNAs, we further examined whether the mechanism is unique to germline cells or common in somatic tissues. We therefore analyzed publicly available eccDNA data from various mouse tissues (*Dillon et al., 2015*). Sequence-based prediction revealed significantly higher nucleosome occupancy probability for ~180 bp and ~360 bp

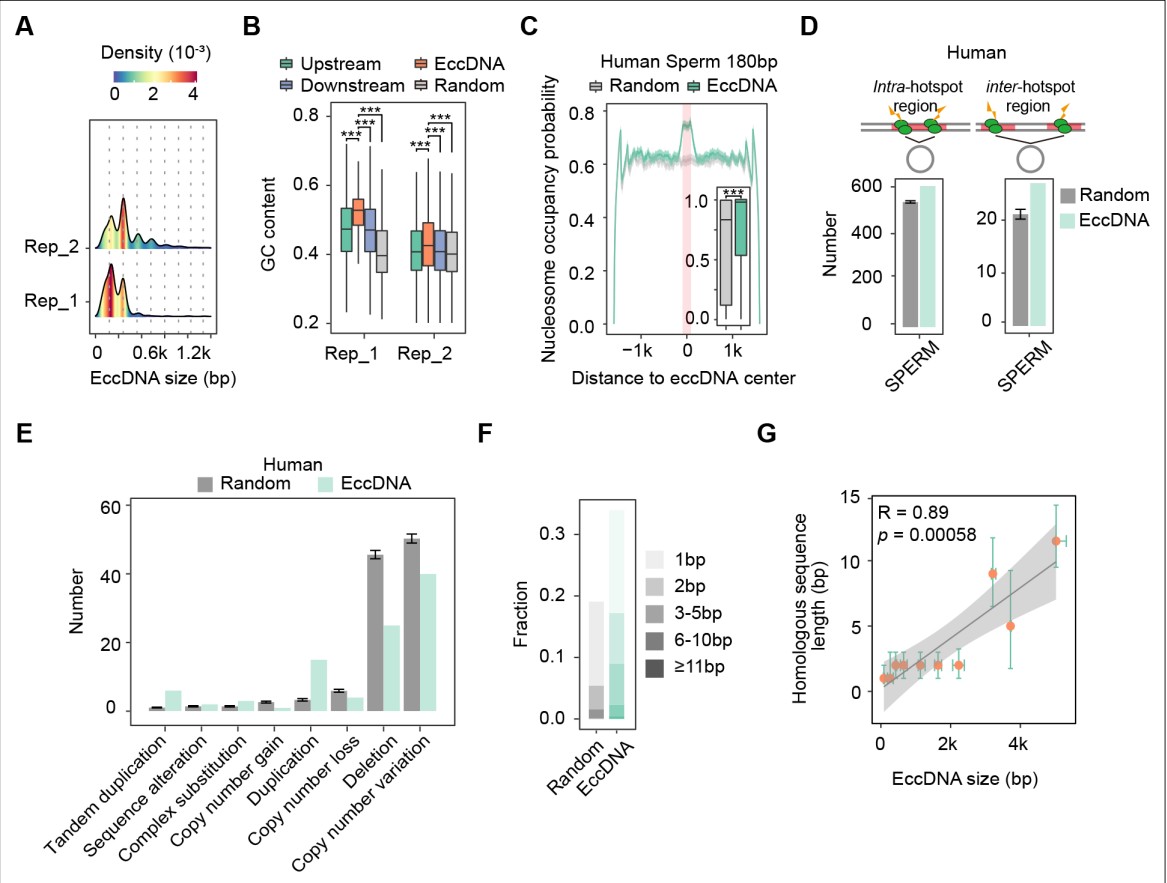

**Figure 5.** The biogenesis mechanism of germline extrachromosomal circular DNAs (eccDNAs) is conserved between human and mouse. (**A**) Size distribution of sperm eccDNAs in two biological replicates. (**B**) GC contents of sperm eccDNAs, regions upstream and downstream of eccDNAs, and randomly selected length-matched control regions. ***Two-sided *Wilcoxon* test p-value<0.001. (**C**) Predicted probability of nucleosome occupancy for eccDNA and randomly selected length-matched control regions (highlighted by red-shaded area), and surrounding regions. Boxplots showing the probability distribution of individual eccDNAs and control regions. ***Two-sided *Wilcoxon* test p-value<0.001. (**D**) Numbers of eccDNAs or randomly selected control regions overlapped with recombination hotspots in human. eccDNAs located completely within a hotspot (*Intra*-), or with both ends overlapped with two different hotspots (*Inter*-) are shown separately. (**E**) Shown are numbers of human sperm eccDNAs or randomly selected control regions having 95% reciprocal overlap with different types of structural variations. (**F**) Percentages of homologous sequences of different lengths (coded by different color saturation levels) are shown for eccDNAs and randomly selected control regions. (**G**) Length of microhomologous sequences as a function of the eccDNA size. Data points are shown as median plus lower (25%) and upper (75%) quartiles. The shaded area is 95% confidence interval of linear regression line. *Pearson* correlation coefficient and two-sided *t*-test p-value are indicated.

The online version of this article includes the following figure supplement(s) for figure 5:

**Figure supplement 1.** Biogenesis mechanism of extrachromosomal circular DNAs (eccDNAs) in mouse somatic tissues and human sperms.

eccDNA regions, suggesting their origin from oligonucleosomal fragments (*Figure 5—figure supplement 1A*). In contrast to simulated controls (~20%), more than 1/3 of eccDNAs had microhomologous sequences, most of which were shorter than 5 bp (*Figure 5—figure supplement 1B*). The remaining 2/3 of eccDNAs had the same sequence motifs between eccDNA starts and sequences following eccDNA ends, and between eccDNA ends and sequences in front of eccDNA starts (*Figure 5—figure supplement 1C*). The genomic distribution of eccDNAs closely matched with that of eccDNAs whose generation was dependent on apoptotic DNA fragmentation (*Figure 5—figure supplement 1D*). Altogether, these results indicate that microhomology-directed ligation of oligonucleosomal fragments in apoptotic cells significantly contributes to eccDNA biogenesis in different mouse tissues.

We next examined whether the mechanism is also conserved in human by analyzing the features of human sperm eccDNAs detected in this study (see 'Materials and methods'). Similarly, human sperm eccDNAs were originated from oligonucleosomal fragmentation as well, as suggested by the pronounced size periodicity of ~180 bp (*Figure 5A*), higher GC content, and nucleosome occupancy

probability (*Figure 5B and C*). We further performed base-level comparison between eccDNAs and meiotic recombination hotspots. Consistent with what we observed in mouse, only about 600 eccDNAs (<0.7% of human sperm eccDNAs) were located in one recombination hotspot or two different hotspots (*Figure 5D*), and only dozens of, or a few hundred eccDNAs in human sperm cells coincided with known genomic deletions (*Figure 5E*). Overall, our analysis disfavors the meiotic recombination to eccDNA biogenesis. Instead, we observed higher frequency of micro-homologous sequences for eccDNAs than simulated control regions (*Figure 5F*). Moreover, large-sized eccDNAs had longer homologous sequences possibly for stable base pairing, as indicated by the strong positive correlation between lengths of homologous sequences and eccDNA sizes (*Figure 5G*). Altogether, these results suggest that microhomology-directed ligation of nucleosome protected DNA fragments is a conserved pathway for germline eccDNA generation in both human and mouse.

## Discussion

The biogenesis mechanisms and biological implications of eccDNAs are big puzzles. Increasingly evidence has emerged to relate eccDNA formation with nucleosome-protected DNA fragments; however, to our knowledge, no study provides direct evidences. The comparison between high number of sperm eccDNAs and retained histones in sperm cells after histone-to-protamine transition (*Torres-Flores and Hernández-Hernández, 2020*) provide a good way to clarify the abovementioned issues. Here, we profiled eccDNA landscape during mouse spermatogenesis and explicitly linked eccDNAs with nucleosome-protected DNA fragments (*Figure 2*). According to these results, we speculate that any oligonucleosomal fragments might have a chance of eccDNA formation as long as microhomologous sequences are present at the ends. In other words, any single cells might generate distinct sets of eccDNAs and thus the theoretical number of eccDNA is extremely large. This might explain why sperm cells, which have a higher starting cell number, have more eccDNAs detected, even though the same amount of eccDNAs (10 ng) for each cell type was used for library preparation and high-coverage sequencing (~150 million reads).

Multiple biogenesis mechanisms of eccDNA have been proposed (*Chiu et al., 2020*; *Dillon et al., 2015*; *Møller et al., 2015*; *Paulsen et al., 2021*; *Sin et al., 2020*); it is however unclear which one plays the dominant role. With our nucleotide-resolution eccDNA breakpoint detection method, we demonstrated that microhomologous sequences are present at boundaries of most eccDNAs (*Figure 4*), suggesting the MMEJ-medicated eccDNA formation as the major mechanism for eccDNA biogenesis. However, rolling circle amplification in Circle-seq protocol preferentially increases the copy numbers of smaller eccDNAs, and our eccDNA detection method relying on uniquely mapped reads might favor the detection of small-sized eccDNAs with homologous sequences. It remains to be determined whether these small-sized eccDNAs with microhomologies are the dominant eccDNA species in the native composition. To examine whether the unexplored size populations of eccDNAs by Circle-seq were also associated with microhomologous sequences, we analyzed eccDNA data generated with long-read sequencing (*Henriksen et al., 2022*) or amplification-free strategies (*Mouakkad-Montoya et al., 2021*). Our sequence feature analyses also revealed the presence of homologous sequences surrounding eccDNA breakpoints (*Figure 5—figure supplement 1E and F*), suggesting the involvement of MMEJ-medicated ligation for large-sized eccDNA as well. We further found the biogenesis mechanism of germline eccDNAs is common to eccDNAs in other tissues, and conserved between human and mouse. However, it remains unclear whether eccDNA generation from healthy cells not undergoing apoptosis is also medicated by MMEJ.

Spontaneous germ cell death has been observed during the normal spermatogenesis (*Liu et al., 2017*; *Shaha et al., 2010*; *Weinbauer et al., 2001*; *Young et al., 2001*). Given the periodic size distribution (*Figure 1C*) and similar genomic features with eccDNAs from apoptotic DNA fragmentation in somatic cells (*Figure 1*, *Figure 4—figure supplement 1*), we reason that mouse germline eccDNAs may be also the germ cell death products. As the final place for sperm maturation and storage, epididymis might contain unhealthy sperm cells undergoing cell death and so DNA fragments, which might additionally contribute to the high eccDNA loads of sperm cells than others. It is possible that failure of histone-to-protamine exchange might account for some sperm cell death. It has been shown that H3.3, generally linked to H3K4me3 marks, plays key roles in modulating TP1 removal and PRM1 incorporation for nucleosome eviction and replacement by protamine (*Wang et al., 2019*), and loss of H3.3 will increase the cell death rate (*Yuen et al., 2014*). H3K36me3 and H3K9me3, whose demethylase

is inhibited by H3.3 (*Udugama et al., 2022*), are also involved in histone replacement (*Wang et al., 2019*), failure of which might underlie the eccDNA generation as well. In line with it, sperm eccDNA regions correspond to depletion of H3.3 variant and H3K4me3, H3K36me3, and H3K9me3 histone modifications (*Figure 2D and E*). This suggests the number of sperm eccDNAs might be highly associated with cell death, and sperm eccDNA may serve as a clinical biomarker for quality assessment of healthy sperms.

It is still debatable whether meiotic recombination is an important source of eccDNA generation. Both positive and negative correlations between eccDNAs and meiotic recombination rates at chromosomal level have been reported previously (*Henriksen et al., 2022*; *Lukaszewicz et al., 2021*). Our results suggest that this discrepancy can be largely reconciled in the light of two eccDNA groups of different sizes. We noted that small-sized and large-sized eccDNAs are preferentially derived from euchromatin and heterochromatins, respectively (*Figure 3*). Since meiotic recombination hotspots are enriched at euchromatin regions (*Lange et al., 2016*), small-sized eccDNAs are therefore positively correlated with recombination rates but the opposite is true for large-sized eccDNAs (*Figure 3—figure supplement 1*). Therefore, the observed correlations between eccDNA density and recombination meiotic rate at chromosomal level are simply indirect reflections of their preferences to different chromatin regions.

We further investigated the overlaps between eccDNAs and meiotic recombination hotspots and structural variations at base level, and found that sperm cell eccDNAs does not seem to be a major source or by-products of de novo deletions, likely because that extensive eccDNA formation might only occur at dead sperm cells that have no chance to be transmitted to the next generation. However, we cannot fully exclude the possibility that some eccDNAs emerging in normal sperm cells and do not affect their viability will be accompanied by de novo deletions or re-integrated into genome to create structural variations. Furthermore, they might stimulate innate immune responses (*Wang et al., 2021a*), serve as extracellular vesicles encoding small RNAs (*Paulsen et al., 2019*), and be taken as biomarkers (*Lv et al., 2022*), which represent exiting research directions.

# Materials and methods

## Isolation of mouse germ cells

Testes from adult C57BL/6 mice were decapsulated, and the seminiferous tubules were torn into small pieces. After incubating in 8 ml PBS containing 1 mg/ml collagenase (Sigma, C5138, St. Louis, MO) and 1 mg/ml hyaluronidase (Sigma, H3506) at 37°C for 6 min, the dispersed seminiferous tubules and cells were incubated at 37°C for 5 min with gentle shaking. Cells were collected by centrifugation at 200 × *g* for 5 min at 4°C, and washed once with PBS, resuspended in 15 ml PBS containing 0.25% Trypsin (Gibco, 25200072) and 1 mg/ml DNase I (Sigma, AMPD1), and incubated at 37°C for 5 min with gentle shaking. Thereafter, cells were collected and washed once with PBS. After filtration through a 70 µm Nylon Cell Strainer (BD Falcon, 352350), the cells were suspended in 30 ml of PBS and stained at 37°C with 5 mg/ml Hoechst 33342 (Thermo Scientific, 62249) for 30 min. Hoechst fluorescence were detected with a 450 nm band-pass filter for blue fluorescent (Hoechst Blue) or a 675 nm band-pass filter for red fluorescent (Hoechst Red). Mouse spermatocytes, round spermatids, and elongating spermatids were sorted by BD FACSAria Fusion (BD Biosciences). Mouse mature spermatozoa were collected form mouse cauda epididymis. The cauda epididymis was quickly cut into pieces and incubated in 1 ml pre-warmed human tubal fluid (HTF) (Millipore, MR-070-D) for 15 min at 37°C, thus allowing the mature spermatozoa to release from the tissue. After filtration through a 40 µm Nylon Cell Strainer (BD Falcon, 352340), mature spermatozoa were collected and washed three times with PBS. About $10^6$ SPA, RST, and EST cells and $10^7$ sperm cells were obtained and used for the eccDNA extraction. All of the animal experiments were performed according to approved institutional animal care and use committee (IACUC) protocols (#08-133) of the Institute of Zoology, Chinese Academy of Sciences.

## Immunofluorescence

Mouse germ cells were spread on glass slides that were air-dried. The slides were fixed in 4% PFA at room temperature for 5 min and washed with PBS three times. After blocking with 5% bovine serum albumin, the primary antibodies (rabbit anti-SYCP3 pAb [Abcam, ab15093, 1:400]; mouse anti-γH2AX

pAb [Millipore, 05-636, 1:400]; mouse anti-sp56 mAb [QED Bioscience, 55101, 1:200]; rabbit Tubulin pAb [ABclonal, AC007, 1:100]) were added to the slides, followed by 14–16 hr incubation at 4°C. After washing the slides with PBS, the secondary antibodies were added, followed by 1.5 hr incubation. The cell nuclei were stained with DAPI for 5 min. Images were observed using a fluorescence microscope SP8 microscope (Leica).

## Human adult sperm sample preparation

The sperm donation candidates in this study were healthy young Chinese men. Each candidate completed a medical examination and extensive medical/social questionnaire to exclude any potential individuals with genetic or major medical problems (such as cardiovascular diseases and sexually transmitted diseases) listed in the Basic Standard and Technical Norms of Human Sperm Bank published by Chinese Ministry of Health. Smokers, drug abusers, and heavy drinkers were also excluded. The rest of the candidates signed a voluntary sperm donation informed consent and agreed to live in Beijing for at least 6 mo. The sperm bank also recorded the candidates' age, date of birth, and date of semen collection. The ethical approval in this study was provided by the Reproductive Study Ethics Committee of Peking University Third Hospital (2017SZ-035). Semen samples were selected through 40% density gradient of PureSperm (Nidacon International, Molndal, Sweden) by centrifugation (500 × g, 30 min) at room temperature and washed with phosphate-buffered saline (PBS) for three times; the obtained spermatozoa were used for the eccDNA extraction.

## Circle-seq

Purification of mouse and human eccDNAs was performed as previously described, with minor modifications (Møller et al., 2018). In brief, samples were resuspended in 500 µl Lysis solution (10 mM Tris pH 7.4, 100 mM NaCl, 1% SDS, 1% Sarkosyl, 150 mM DTT) with 10 µl Proteinase K (Thermo Scientific, EO0491) and incubated overnight at 55°C. After cell lysis, phenol:chloroform:isoamyl alcohol was added and mixed, and centrifuged at 13,000 × g for 15 min at 4°C. The supernatant was moved to a new tube, incubated with 500 µl isopropanol at room temperature for 10 min, and centrifuged at 13,000 × g for 15 min at 4°C. The resulted DNA pellet was washed with 1 ml 70% ethanol, treated with alkaline to separate chromosomal DNA from eccDNAs by rapid DNA denaturing–renaturing, followed by column chromatography on an ion exchange membrane column (TIANprep Mini Plasmid Kit, DP103). Column-bound DNA was eluted in TE buffer (10 mM Tris-Cl, pH 8.0; 1 mM EDTA, pH 8.0) and treated with AsiSI (NEB, R0630S) and PacI (NEB, R0547S) endonucleases at 37°C for genomic DNA and mtDNA fragmentation. The remaining linear DNA was treated by exonuclease (Plasmid-Safe ATP-dependent DNase, Lucigen, E3101K) at 37°C for 1 wk, during which additional ATP and DNase was added every 24 hr (30 units per day) according to the manufacturer's protocol (Plasmid-Safe ATP-dependent DNase, Lucigen, E3101K). The eccDNA samples were cleaned by phenol:chloroform:isoamyl alcohol once, followed by ethanol precipitation. EccDNAs were then amplified by phi29 polymerase (NEB, M0269L) at 30°C for 16 hr. Phi29-amplified DNA was cleaned by phenol:chloroform:isoamyl alcohol once, followed by ethanol precipitation. The DNA samples were sonicated to a set size of 250 bp with an M220 Focused-ultra sonicator (Covaris, Woburn). Sequencing libraries were generated with NEBNext Ultra II DNA Library Prep Kit for Illumina (NEB, E7645S), according to the manufacturer's instructions. Then, 10 ng of eccDNA samples were used for library preparation. NEBNext Multiplex Oligos for Illumina (Set 1, NEB #E7600) were used for PCR amplification of adaptor-ligated DNA. Libraries were purified with SPRIselect Reagent Kit (Beckman Coulter, Inc #B23317). Paired-end 150 bp sequencing was performed on Illumina NovaSeq 6000 System.

## Quality control of Circle-seq and eccDNA validation

Exogenous circular DNA (pUC19) and mtDNA were measured by qPCR in a QuantStudio 6 Flex Real-Time PCR System, following the manufacturer's protocol. The primer probes used for qPCR included pUC19 forward: 5'-AGC GAA CGA CCT ACA CCG AAC-3', pUC19 reverse: 5'-CTC AAG TCA GAG GTG GCG AAA C-3'; MTND2 forward: 5'-AAC AAA CGG TCT TAT CCT TAA CAT AAC A-3', MTND2 reverse: 5'-TGG GAT CCC TTG AGT TAC TTC TG-3'; H19 forward: 5'-GTA CCC ACC TGT CGT CC-3', H19 reverse: 5'-GTC CAC GAG ACC AAT GAC TG-3'. EccDNA validation was performed by outward PCR in genomic DNA and eccDNA extractions. The primer probes used for eccDNA validation included Clone 1 (chr1:73132172–73132640) in-forward: 5'-TTT TCC TGG AGC ACA CTA GC-3';

Clone 1 in-reverse: 5′-CAT GCT AAA CAA AGC ATG TCA C-3′; Clone 1 out-reverse: 5′-CAA CTG ACA CCA ACC ACA TC-3′; Clone 2 (chr1:120058476–120058826) in-forward: 5′-CCT GCC ACT GCT CTG CAT TC-3′; Clone 2 in-reverse: 5′-AGA TGC AAT AGG ACC AGG ATG-3′; Clone 2 out-reverse: 5′-GCC CAG AGC AGA ATC CAA AG-3′; Clone 3 (chr13:95405916–95406337) in-forward: 5′-GGT CAC ACA TGC AAA TGT CC-3′; Clone 3 in-reverse: 5′-AAC ATA CCT GAG ACC CTA GG-3′; Clone 3 out-reverse: 5′-TTC CCA CAG CTA TGC TCA GC-3′.

## EccDNA detection

We developed a nucleotide-resolution eccDNA detection pipeline on the basis of previous efforts (*Figure 4—figure supplement 1B*; *Kumar et al., 2017*; *Møller et al., 2018*). Briefly, SeqPrep (*St John, 2016*; https://github.com/jstjohn/SeqPrep) was used to trim adapter sequences and merge the overlapping paired-end reads into singleton long reads, followed by reads mapping to GRCm38 reference genome using BWA MEM version 0.7.17-r1188 (*Li and Durbin, 2009*). Samblaster version 0.1.26 (*Faust and Hall, 2014*) or an in-house Perl script was used to remove PCR duplicates and separate alignments into split reads, discordant and concordant reads. Candidate eccDNAs were firstly identified based on split reads (high-confidence ones). If the total length of two sub-alignments of split reads exceeded the read length, homologous sequences were searched. When homologous sequences were found, we recorded the coordinates of the leftmost form of eccDNA and an offset corresponding to the length of homologous sequences to represent all possible eccDNA variants. Potential split reads that failed to be mapped as split reads in the first place (low-confidence ones) as well as discordant reads were identified and counted using in-house Perl scripts. The average coverages (in terms of RPK) for candidate eccDNAs and surrounding regions were then calculated based on all different types of reads. Any eccDNA supported by at least two high-confidence split reads or discordant reads, with its 95% region covered by at least one read, and with its average coverage twice of that of its surrounding region, is considered as a high-confidence eccDNA.

## Evaluation of our eccDNA detection method

We randomly selected 1000 eccDNAs containing ≥2 bp of microhomology sequences from this study. Ten copies of each eccDNA sequence were concatenated to mimic the product of rolling amplification process. We then randomly extracted 50 fragments ranging from 250 to 350 bp from each concatenated sequence to mimic the sonication process and generated 150 bp paired-end reads from each fragment. We applied Circle_finder, Circle_Map, ecc_finder, ECCsplorer, and our own method to identify eccDNAs from the simulated paired-end reads. Detected eccDNAs with at least 95% reciprocal similarity with one of the 1000 eccDNAs were considered positive hits. We considered eccDNA boundaries to be correctly assigned only if they had the same start and end coordinates.

## EccDNA characterization

Gene structure annotations in mouse and human are based on Ensembl GRCm38 Release 102 and Ensembl GRCh38 Release 104, respectively. Repeat elements are annotated by RepeatMasker database. Random region generation, sequence composition calculation, and overlapping region determination were all done with Bedtools version 2.30.0 (*Quinlan and Hall, 2010*). The nucleosome occupancy probability of the eccDNAs and the surrounding regions was predicted using predNuPoP function of NuPoP version 2.2.0 R package (*Xi et al., 2010*). Motif analysis and visualization were done by Two Sample Logo web server (*Vacic et al., 2006*). Public ChIP-seq datasets (*Jung et al., 2019*; *Jung et al., 2017*; *Singh and Parte, 2021*) were aligned to GRCm38 reference genome using Bowtie, and sorted and indexed using SAMtools version 1.7 (*Danecek et al., 2021*). Picard MarkDuplicates version 2.18.14 was used to remove PCR duplicates. Peak calling was done with Macs2 version 2.2.7.1 (*Zhang et al., 2008*). Coverage file was generated from BAM file and visualized using deeptools version 3.5.1 (*Ramírez et al., 2016*).

## Acknowledgements

The authors are grateful to the members of the J-YC, WL, and CL labs for cooperation, reagent sharing, and insightful discussions during the course of this investigation. This study is supported by the MOST (2021YFF1201500 to J-YC and 2022FYC2702600 to HJ) and NSFC (32170653 to J-YC, 81925015 and 32230029 to WL, and 32270898 to CL).

# Additional information

## Funding

| Funder | Grant reference number | Author |
|--------|------------------------|--------|
| Ministry of Science and Technology (China) | 2021YFF1201500 | Jia-Yu Chen |
| National Natural Science Foundation of China | 32170653 | Jia-Yu Chen |
| National Natural Science Foundation of China | 81925015 | Wei Li |
| National Natural Science Foundation of China | 32230029 | Wei Li |
| National Natural Science Foundation of China | 32270898 | Chao Liu |
| Ministry of Science and Technology (China) | 2022YFC2702600 | Hui Jiang |

The funders had no role in study design, data collection and interpretation, or the decision to submit the work for publication.

## Author contributions

Jun Hu, Data curation, Software, Formal analysis, Investigation, Visualization, Methodology; Zhe Zhang, Resources, Validation, Investigation, Methodology; Sai Xiao, Yalei Cao, Yinghong Chen, Jiaming Weng, Validation, Investigation, Methodology; Hui Jiang, Resources, Supervision, Project administration; Wei Li, Conceptualization, Resources, Supervision, Funding acquisition, Project administration; Jia-Yu Chen, Conceptualization, Resources, Software, Supervision, Funding acquisition, Investigation, Methodology, Writing – original draft, Project administration, Writing – review and editing; Chao Liu, Conceptualization, Resources, Formal analysis, Supervision, Funding acquisition, Validation, Investigation, Visualization, Methodology, Writing – original draft, Project administration, Writing – review and editing

## Author ORCIDs

Jia-Yu Chen ⓘ https://orcid.org/0000-0001-9449-9321

## Ethics

Human subjects: The sperm donation candidates in the present study were healthy young Chinese men. Each candidate completed a medical examination and extensive medical/social questionnaire to exclude any potential individuals with genetic or major medical problems (such as cardiovascular diseases and sexually transmitted diseases) listed in the Basic Standard and Technical Norms of Human Sperm Bank published by Chinese Ministry of Health. Smokers, drug abusers, and heavy drinkers were also excluded. The rest of the candidates signed a voluntary sperm donation informed consent and agreed to live in Beijing for at least 6 months. The sperm bank also recorded the candidates' age, date of birth, and date of semen collection. The ethical approval in this study were provided by the Reproductive Study Ethics Committee of Peking University Third Hospital (2017SZ-035).

All of the animal experiments were performed according to approved institutional animal care and use committee (IACUC) protocols (#08-133) of the Institute of Zoology, Chinese Academy of Sciences. The protocol was approved by the committee on the ethics of animal experiments of the Institute of Zoology, Chinese Academy of Sciences (Permit Number: IOZ20190038).

Joint Public Review: https://doi.org/10.7554/eLife.87115.3.sa1
Author Response https://doi.org/10.7554/eLife.87115.3.sa2

# Additional files

## Supplementary files

• MDAR checklist

## Data availability

The raw sequencing data reported in this study have been deposited in the Genome Sequence Archive (*Chen et al., 2021*) in the National Genomics Data Center (*CNCB-NGDC Members and Partners, 2022*; GSA accession numbers: CRA008015 and HRA002957) that are publicly accessible at https://ngdc.cncb.ac.cn/gsa. The eccDNA mapping and detection workflows (v1.0) are available at https://github.com/NjuChenlab/eccDNA_detector_tools (*NjuChenlab, 2022*).

The following datasets were generated:

| Author(s) | Year | Dataset title | Dataset URL | Database and Identifier |
|---|---|---|---|---|
| Hu J, Zhang Z, Xiao S, Cao Y, Chen Y, Weng J, Jiang H, Li W, Chen J-Y, Liu C | 2022 | Widespread Microhomology-Mediated Circular DNA Formation from Oligonucleosomal DNA Fragmentation During Mouse Spermatogenesis | https://ngdc.cncb.ac.cn/search/?dbId=&q=CRA008015 | Genome Sequence Archive, CRA008015 |
| Hu J, Zhang Z, Xiao S, Cao Y, Chen Y, Weng J, Jiang H, Li W, Chen J-Y, Liu C | 2022 | Widespread eccDNA formation in human sperm | https://ngdc.cncb.ac.cn/gsa-human/browse/HRA002957 | Genome Sequence Archive, HRA002957 |

The following previously published datasets were used:

| Author(s) | Year | Dataset title | Dataset URL | Database and Identifier |
|---|---|---|---|---|
| Jung YH, Kremsky I, Gold HB, Rowley MJ, Punyawai K, Buonanotte A, Lyu X, Bixler BJ, Chan AWS, Corces VG | 2019 | Maintenance of CTCF and transcription factor-mediated interactions from gametes to the early mouse embryo | https://www.ncbi.nlm.nih.gov/geo/query/acc.cgi?acc=GSE116857 | NCBI Gene Expression Omnibus, GSE116857 |
| Jung YH, Sauria MEG, Lyu X, Cheema MS, Ausio J, Taylor J, Corces VG | 2017 | Mammalian sperm exhibit epigenetic priming of embryonic and adult regulatory landscapes (ChIP-Seq) | https://www.ncbi.nlm.nih.gov/geo/query/acc.cgi?acc=GSE79227 | NCBI Gene Expression Omnibus, GSE79227 |
| Singh I, Parte P | 2021 | Epigenetic landscape of murine testicular histone variants TH2A and TH2B in sperm | https://www.ncbi.nlm.nih.gov/geo/query/acc.cgi?acc=GSE181921 | NCBI Gene Expression Omnibus, GSE181921 |
| Alleva B, Brick K, Pratto F, Huang M, Camerini-Otero RD | 2021 | Cataloging human PRDM9 variability utilizing long-read sequencing technologies reveals PRDM9 population-specificity and two distinct groupings of related alleles | https://www.ncbi.nlm.nih.gov/geo/query/acc.cgi?acc=GSE166483 | NCBI Gene Expression Omnibus, GSE166483 |

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
