## [Editor Report · eLife assessment]

This study provides **important** information on the biogenesis of eccDNAs during spermatogenesis. The data presented are **solid** and supportive of the concussion that eccDNAs in spermatogenic cells are not derived from miotic recombination hotspots but rather represent oligonucleosomal DNA fragments from apoptotic male germ cells, whose ends are ligated through microhomology-mediated end-joining. This work is of interest to researchers working on germ cell biology and cancer biology.

---

## [Referee Report · Joint Public Review]

This study investigated the mechanisms and biological processes associated with eccDNA generation in germline cells. They enriched eccDNA from cells at each step of spermatogenesis in mouse as well as human sperm using a commonly-used method to enrich small eccDNA: column purification, exonuclease digestion, rolling circle amplification, followed by short-read Illumina sequencing. From the fragment size analyses, dominant sizes were shown to be those protected by mono- or di-nucleosomes. The authors developed a computational pipeline to investigate eccDNA breakpoints in detail from split reads and reported a prevalence of a microhomology-mediated mechanism. Features of small germline eccDNA closely matched with small eccDNA generated by apoptosis, suggesting apoptotic germline cells as a major source.

Combined with analyses of publically available data from mouse tissues, the study established a strong link between small eccDNA and DNA fragments protected by mono-or di-nucleosomes. The rigorous investigation of microhomologies revealed that eccDNA sizes correlated with the lengths of microhomology in spermatogonial cells. Small eccDNA tends to originate from euchromatic regions, while longer eccDNA is derived from heterochromatic regions. These are novel findings.

The authors repeatedly stated the rare association between eccDNA and recombination hotspots. The argument was backed by (1) the abundance of eccDNA coming from dead cells, and (2) the small number of eccDNA from SPA cells undergoing miosis. The argument seems to have a point; however, the observation that the authors recovered hundreds of eccDNA at recombination hotspots may indicate that miotic recombination is a significant source of eccDNA. Because of the bulk isolation of eccDNA, those eccDNAs were outnumbered by the abundant eccDNA coming from apoptotic cell death. Indeed, eccDNAs from recombination hotspots are slightly more than random in all cell types (Fig. 4A).

Related to this issue, the dominance of both 180- and 360-bp fragments in most mouse tissues put the single 180-bp peaks of SPA, RST, and EST eccDNA in a peculiar position. Fewer numbers of these cells were used for eccDNA isolation than sperm cells, resulting in fewer eccDNA in these cell types, despite the same amount (10ng) of DNA input for rolling circle amplification. There might be a technical issue behind the peculiar observation, which is understandable given the challenging nature of isolating pure cell populations.

---

## [Author Response]

The following is the authors’ response to the original reviews.

Thank you for submitting your article "Microhomology-Mediated Circular DNA Formation from Oligonucleosomal Fragments During Spermatogenesis" for consideration by eLife. Your article has been reviewed by 2 peer reviewers, and the assessment has been overseen by a Reviewing Editor and Diane Harper as the Senior Editor.

**eLife assessment**
This study provides valuable information on the biogenesis of eccDNAs during spermatogenesis, i.e., eccDNAs in spermatogenic cells are not derived from miotic recombination hotspots but represent oligonucleosomal DNA fragments from apoptotic male germ cells, whose ends are ligated through microhomology-mediated end-joining. The study is currently incomplete because the method of bioinformatics needs more details and data interpretation should take the amplification bias into consideration.

We highly appreciate the positive assessment of our manuscript. Following the insightful suggestions by editors and two reviewers, we have fully addressed two major concerns, i.e., the missing of method detail and the biased data interpretation.

First, to provide the detail of our bioinformatics methods, (i) We have illustrated the principle and steps of our eccDNA detection method by Figure 4C and Figure 4-figure supplement 2B, and submitted our source codes to GitHub (website); (ii) We compared the performance of our methods in comparison with four established bioinformatics tools on both simulated and real datasets, and revealed that it has comparable sensitivity and specificity (Figure 4—figure supplement 2C and E), and much higher accuracy on the assignment of eccDNA boundaries (Figure 4—figure supplement 2A, D and F); and (iii) we have added more description to help readers to better understand our method (see Methods – eccDNA Detection).

Second, the amplification bias is indeed a problem of Circle-seq. Following editors’ and Reviewer #1’s insightful suggestions, we analyzed other datasets generated by amplification-free strategies (Mouakkad-Montoya et al., PNAS, 2021) and long-read sequencing (Henriksen et al., Mol Cell, 2022). We identified the presence of homologous sequences surrounding eccDNA breakpoints in both datasets (Figure 5-figure supplement 1E and F), suggesting the involvement of MMEJ-medicated ligation for the unexplored size populations of eccDNAs by Circle-seq as well. We have discussed this point and added one section to remind readers of the limitations of rolling-circle amplification-based Circle-seq (the 2nd paragraph of Discussion section).

For your and reviewers’ convenience, all changes in the revised manuscript have been marked in red. We hope the modified manuscript addresses your and the reviewers’ concerns satisfactorily and is suitable for publication in eLife now.

**Reviewer #1 (Public Review):**
This study aims to address the mechanism of eccDNA generation during spermatogenesis in mice. Previous efforts for cataloging eccDNA in mammalian germ cells have provided inconclusive results, particularly in the correlation between meiotic recombination and the generation of eccDNA. The authors employed an established approach (Circle-seq) to enrich and amplify eccDNA for sequencing analyses and reported that sperm eccDNA is not associated with miotic recombination hotspots. Rather, the authors reported that eccDNAs are widespread, and oligonucleosomal DNA fragments from sperm undergoing apoptosis, with the ligation of DNA ends by microhomology-mediated end-joining, would be a major source of eccDNA.The strength of the study includes evaluating the eccDNA contents not only in sperm but also from earlier stages of cells in spermatogenesis. The differences in eccDNA size peaks between sperm and other progenitors, in particular, the unique peak in sperm around 360 bp, are intriguing. Results from sequencing data analysis were presented elegantly.

We are grateful to Reviewer #1 for his or her recognition of the strength of this study.

I also have critiques. First, the lack of eccDNA quality control step is a concern. Previous studies employed electron microscopy to ensure that DNA species are mostly circular before rolling-circle amplification. Phi29 polymerase is widely used for DNA amplification, including whole genome amplification of linear chromosomal DNA. Phi29 polymerase has a high processivity and strand displacement activity. When those activities occur within a molecule, it creates circular DNA from linear DNA in vitro. In vitro-created eccDNA from linear DNA would be randomly distributed in the genome, which may explain the low incidence of common eccDNA between replicates. Therefore, it will be crucial to show that DNA prior to amplification is dominantly circular. Electron microscopy would be challenging for the study because the relatively small number of cells were processed to enrich eccDNA. An alternative method for quality controls includes spiking samples with linear and circular exogenous DNA and measuring the ratios of circular/linear control DNA before and after column purification/exonuclease digestion. eccDNA isolation procedures can be validated by a very high circular/linear control DNA ratio.

We greatly appreciate Reviewer #1's valuable suggestions. We have introduced an exogenous circular DNA (pUC19) into our samples and measured its abundance relative to a linear DNA locus (H19 gene) before and after eccDNA isolation procedures according to Reviewer #1's suggestion. As anticipated, we observed significant enrichment of pUC19 following eccDNA isolation (new Figure 1-figure supplement 2A). These results affirm the high selectivity of our protocol in enriching eccDNAs.

Another critique is regarding the limitation of the study. It is important to remind the readers of the limitations of the study. As the authors mentioned, rolling circle amplification preferentially increases the copy numbers of smaller eccDNA. Therefore, the native composition of eccDNA is skewed. In addition, the candidate eccDNAs are identified by split reads or discordant read pairs. The details of the mapping process are unclear from the methods, but such a method would require reads with high mapping quality; the identification of eccDNA is expected to require sequencing reads that are mapped to genomic locations uniquely with high confidence, and reads mapped to more than one genomic location, such as highly similar repeat sequences or duplications, are eliminated. Such identification criteria would favor eccDNA formed by little or no homology at the junction sequences, and eliminate eccDNA formed by long homologies at the ends, such as eccDNA formed exclusively by satellite DNA. Therefore, it is not surprising that the authors found the dominance of microhomology-mediated eccDNA. It remains to be determined whether small eccDNA with microhomologies are the dominant species of eccDNA in the native composition. In this regard, it is noted that similar procedures of eccDNA enrichment (column purification, exonuclease digestion, and rolling circle amplification ) revealed variable sizes and characteristics of eccDNA in sperm (human from Henriksen et al. or mice from this study), dependent on the methods of sequencing (long-read or short-read sequencing). Considering these limitations, the last sentence of the introduction, "We conclude that germline eccDNAs are formed largely by microhomology mediated ligation of nucleosome protected fragments, and barely contribute to de novo genomic deletions at meiotic recombination hotspots" needs to be revised.

We thank Reviewer #1 for bringing attention to the limitations of the study. Since rolling circle amplification preferentially increases the copy numbers of smaller eccDNA, the exact size distribution of eccDNA in native composition is yet to be determined. As pointed out by Reviewer #1, our mapping and eccDNA detection processes might indeed introduce some biases since we only focused on uniquely-mapped reads. We have addressed and incorporated Reviewer #1’s perspectives in our revised manuscript, as detailed in the 2nd paragraph of Discussion section.

Despite these limitations, microhomology mediated ligation of DNA fragments seems to be the major mechanism of eccDNA biogenesis nonetheless. We analyzed eccDNA datasets generated through long-read sequencing (Henriksen et al., Mol Cell, 2022) or amplification-free strategies (Mouakkad-Montoya et al., PNAS, 2021). Although these eccDNAs represented size populations that were largely missed by this study, our sequence feature analyses also revealed the presence of homologous sequences surrounding eccDNA breakpoints, as depicted in the newly added Figure 5-figure supplement 1E and F. Considering that we could not totally overcome these biases in this study, we have toned down some statements and revised the last sentence of the introduction as follows: “We conclude that germline eccDNAs are likely formed by microhomology mediated ligation of nucleosome-protected fragments, and barely contribute to de novo genomic deletions at meiotic recombination hotspots.”

Small eccDNA (microDNA) data from various mouse tissues are available from the study by Dillion et al., (Cell Reports 2015). Authors are encouraged to examine whether the notable findings in this study (oligonucleosomal-sized eccDNA peaks and the association with apoptotic cell death) are unique to sperm or common in the eccDNA from other tissues.

We are thankful to Reviewer #1 for this suggestion. We analyzed eccDNA data from various mouse tissues (Dillion et al., Cell Rep, 2015) to see whether our findings are unique to sperm or common for other tissues. Sequence-based prediction revealed significantly higher nucleosome occupancy probability for ~180 bp and ~360bp eccDNA regions, suggesting their origin from oligonucleosomal fragments (Figure 5-figure supplement 1A). In contrast to simulated controls (~20%), more than 1/3 of eccDNAs had microhomologous sequences, most of which were shorter than 5bp (Figure 5-figure supplement 1B). The remaining 2/3 of eccDNAs had the same sequence motifs between eccDNA starts and sequences following eccDNA ends, and between eccDNA ends and sequences in front of eccDNA starts (Figure 5-figure supplement 1C). The genomic distribution of eccDNAs closely matched with that of eccDNAs whose generation was dependent on apoptotic DNA fragmentation (new Figure 5-figure supplement 1D). Altogether, these results indicate microhomology directed ligation of oligonucleosomal fragments in apoptotic cells significantly contributes to eccDNA biogenesis in different mouse tissues. We have described this part in the revised manuscript (see the last 2nd paragraph of Results section).

**Reviewer #2 (Public Review):**
This study presents a useful investigation of eccDNAs in spermatogenesis of mouse. It provides evidence about the biogenesis of eccDNAs and suggests that eccDNAs are derived from oligonucleosmal DNA fragmentation during apoptosis by MMEJ and may not be the direct products of germline deletions. However, the method of data analyses were not fully described and data analysis is incomplete. It provides additional observations about the eccDNA biogenesis and can be used as a starting point for functional studies of eccDNA in sperms. However, many aspects about data analyses and data interpretations need to be improved.

We thank Reviewer #2 for his or her critical reading. We have provided more method details, performed additional analyses and made some clarifications in our revised manuscript (see below).

• Most of the conclusions made by the work are only based on the bioinformatics analyses, the validation of these foundlings using other method (biochemistry/molecular biology method) are missing. For example, no QC results presented for the eccDNA purification, which may show whether contaminates such as linear DNA or mitochondria DNA have been fully removed. Additionally, it is also helpful to use simple PCR to test the existence of identified eccDNAs in sperm or other samples to validate the specificity of the Circle-seq method.

Following both this Reviewer’s and Reviewer #1’s suggestions, we performed quality control of eccDNA purification. First, we introduced an exogenous circular DNA (pUC19) into our samples and measured its abundance relative to a linear DNA locus (H19 gene) before and after eccDNA isolation procedures. As anticipated, we observed significant enrichment of pUC19 following eccDNA isolation (Figure 1-figure supplement 2A). Second, mitochondria DNA is supposed to be cleaved into linear DNA by PacI and degraded by exonuclease. As expected, the abundance of mitochondria DNA significantly decreased after eccDNA isolation procedures (Figure 1-figure supplement 2B). Third, we performed PCR using outward primers and validated three randomly-selected eccDNAs (Figure 1-figure supplement 2C).

• The reliability of the data analysis methods is uncertain, as the authors constructed and utilized their own pipeline to identify eccDNAs, despite the availability of established bioinformatics tools such as ECCsplorer, eccFinder, and Amplicon Architect. Moreover, the lack of validation of the pipeline using either ground truth datasets or simulation data raises concerns about its accuracy. Additionally, the methodology employed for identifying eccDNA that encompasses multiple gene loci remains unclear.

We thank Reviewer 2 for pointing out this problem. In the original version of our manuscript, focusing on one eccDNA dataset generated in this study, we have compared the performance between our method and established methods for identification of eccDNA regions, such as Circle_finder, Circle_Map and ecc_finder. Our method has comparable sensitivity and specificity with existing methods, especially Circle_finder and Circle_Map (original Figure 4—figure supplement 2C). We also used one specific genomic region to show that existing methods identified the same eccDNA regions but misassigned the eccDNA boundaries (original Figure 4—figure supplement 2A). In the revised manuscript, we have further included ECCsplorer for comparison. Since Amplicon Architect is more specifically designed for detection of ecDNAs, it was not included in our comparison. Following Reviewer #2’ suggestions, we simulated paired-end reads derived from a set of eccDNAs with homologous sequences around breakpoints and employed all methods for eccDNA identification. In total, 97.9%, 97.9%, 97.4%, 95.3% and 91.1% eccDNA regions could be detected by our method, Circle_Map, Circle_finder, ecc_finder and ECCsplorer, respectively (Figure 4—figure supplement 2C). This result suggest that our method has comparable performance in detecting eccDNA regions. However, only our method could faithfully assign breakpoints with 97.4% accuracy, in contrast to no more than 15% by other methods (Figure 4—figure supplement 2D).

As pointed out by Reviewer #2, similar to ECCsplorer, Circle_finder, Circle_Map and ecc_finder, our method fails to identity eccDNAs that encompass multiple gene loci. We have reminded readers of this limitation in our revised manuscript. Besides the schematic workflow (Figure 4—figure supplement 2B), we have included more method details to help readers better understand how our method works (see Methods – eccDNA Detection).

• Although the author stated that previous studies utilizing short-read sequencing technologies may have incorrectly annotated eccDNA breakpoints, this claim requires careful scrutiny and supporting evidence, which was not provided in the manuscript.

Following this Reviewer’s suggestions, we conducted a systematic evaluation of the performance of various existing methods, namely Circle_finder, Circle_Map, ECCsplorer and ecc_finder, for eccDNA breakpoint annotation.

First, we simulated paired-end reads derived from a set of eccDNAs with homologous sequences around breakpoints and employed all different methods for eccDNA identification. As expected, our method could correctly assign breakpoints for 97.4% eccDNAs (Figure 4—figure supplement 2D), in contrast to no more than 15% by other methods (Figure 4—figure supplement 2D).

Second, we examined the performance of all methods on one dataset generated in this study. Our method detected 59,680, 54,898, 32,993 and 22,019 eccDNAs with homologous sequences that were also detected by Circle_finder, Circle_Map, ECCsplorer and ecc_finder, respectively. Remarkably, we observed that at least 60% of breakpoints were misannotated by the existing methods (Figure 4—figure supplement 2F).

We have included an example in Figure 4—figure supplement 2A, where all existing methods incorrectly annotated the eccDNA breakpoints when homologous sequences were present. These results highlight the advantage of our method over existing methods in accurately annotating eccDNA breakpoints in the presence of homologous sequences.

• The similarity between the eccDNA profiles of human and mouse sperm remains uncertain, and therefore, analyses of human eccDNA data and comparisons between the two are necessary if the authors claim that their findings of widespread eccDNA formation in mouse spermatogenesis extend to human sperms.

Our Fig. 5 have shown that human sperm eccDNAs are originated from oligonucleosomal fragmentation (Fig. 5A-C), not associated with meiotic recombination hotspots (Fig. 5D and E) but formed by microhomology directed ligation (Fig. 5F and G). These findings are consistent with what we observed in mouse sperm eccDNAs. To further substantiate our findings, we analyzed an additional eccDNA dataset from human sperms generated by long-read sequencing (Henriksen et al., Mol Cell, 2022). Although predominantly composed of large-sized eccDNAs, the analysis of sequence features also indicated their association with microhomology directed ligation (Figure 5-figure supplement 1E). Overall, the eccDNA profiles in human and mouse sperm exhibit notable similarities.

**Reviewer #1 (Recommendations For The Authors):**
In the last sentence of the abstract, the authors stated, "provide a potential new way for quality assessment of sperms." There is no basis for the claim in the abstract. The authors need to mention the association of eccDNA with apoptosis somewhere to claim it.

We have revised the Abstract as suggested.

Some of the references need to be clarified. For example, Coquelle et al., 2002 described the BFB cycles and common fragile sites, but the report does not seem to be relevant to eccDNA. Mouakkad-Montoya et al., 2021 enriched eccDNA without rolling-circle amplification.

Thanks for pointing this out. We cited Coquelle et al., 2002 to list known biogenesis mechanisms for ecDNAs but not eccDNAs. We have deleted Mouakkad-Montoya et al., 2021 in our revised manuscript, as it did not involve rolling-circle amplification.

**Reviewer #2 (Recommendations For The Authors):**
• It is not clear why the authors took 3000bp as the cutoff to divide eccDNAs into short and long categories. How many long eccDNAs in these samples?

Henriksen et al identified size range of sperm eccDNAs as ~3–50 kb. We therefore used 3kb as an arbitrary cutoff to better compare two different eccDNA populations with those reported by Henriksen et al. SPA, RST, EST and sperm cells have 278, 609, 373 and 691 eccDNAs respectively that are longer than 3000bp. We have clarified this in the revised manuscript.

• In figure 2D,2E, what is the zero point in the heatmaps? The 5', 3' end or center of eccDNA? Please make it clear in figure and main text.

The zero point represents the center of eccDNA regions. We have clarified this in the revised manuscript.

• In line 245, the author mentioned that "periodic distribution of nucleosomes was observed for ~360bp eccDNAs but not for ~180bp ones, indicating that eccDNAs from di-nucleosomes but not mono-nucleosomes preferentially originate from well-positioned nucleosome arrays (Figure 2E)". Please explain how to make the conclusion from the Figure 2E?

Taking the H3K27me3-marked nucleosome as an example, vertical stripes were distributed every ~180bp for ~360bp eccDNAs, as shown by heatmap (more evident if in an enlarged view), and periodic signal distribution was apparent for ~360bp eccDNAs (Figure 2E), as shown by meta-gene analysis on top of heatmap (Figure 2B). However, such pattern was not observed for ~180bp eccDNAs. Similar results could also be observed for nucleosomes marked with other histone variants and histone modifications (H3, H3K27ac, H3K4me1, H3K9ac, H3K36me3, H3K9me3 in Figure 2E). Thus, eccDNAs from di-nucleosomes but not mono-nucleosomes preferentially originate from well-positioned nucleosome arrays in sperm.

• In line 261, the author mentioned: "the large-sized sperm eccDNAs detected in this study also displayed weak but apparent negative correlation with gene density and Alu elements (Figure 3C and D)". However, the data didn't show the "apparent negative correlation", as only one or two data points may support this conclusion and the p-values are not even close to 0.05.

Many thanks for pointing this out. We have toned down this statement as “the large-sized sperm eccDNAs detected in this study displayed a weak negative correlation with gene density or Alu elements (Figure 3C and D)”.

• The enrichment of both active (H3K27ac, H3K9ac) and repressive (H3K9me3) histone markers in the original loci of eccDNA poses an intriguing question: how can this seemingly contradictory pattern be explained? In the H3K9me3 heatmap, the average level of H3K9me3 in eccDNA is lower than control's, how to interpret the result?

We found that small-sized eccDNAs were more enriched at H3K27ac-marked euchromatin regions (Figure 2C-E and 3A), while large-sized ones were more enriched at H3K9me3-marked heterochromatin regions (Figure 3A). This is probably because heterochromatin regions are too condensed to be fragmented into smaller pieces for small-sized eccDNA formation, in comparison with euchromatin regions. We have included this information in our revised manuscript.

H3K9me3 histone marks are enriched at repeat sequences that are widely distributed within the mouse genome. Moreover, the H3K9me3 ChIP-seq dataset we analyzed in this study had the highest number of ChIP-seq peaks, compared to ChIP-seq datasets of other histone modifications. Thus, even random control would probably have stronger ChIP-seq signals than small-sized eccDNAs (e.g., ~180bp or ~360bp eccDNAs) that were preferentially generated from active regions.